# *Pinus sylvestris* L. and *Syzygium aromaticum* (L.) Merr. & L. M. Perry Essential Oils Inhibit Endotoxin-Induced Airway Hyperreactivity despite Aggravated Inflammatory Mechanisms in Mice

**DOI:** 10.3390/molecules27123868

**Published:** 2022-06-16

**Authors:** Eszter Csikós, Kata Csekő, Ágnes Kemény, Lilla Draskóczi, László Kereskai, Béla Kocsis, Andrea Böszörményi, Zsuzsanna Helyes, Györgyi Horváth

**Affiliations:** 1Department of Pharmacognosy, Faculty of Pharmacy, University of Pecs, H-7624 Pecs, Hungary; csikos.eszter@gytk.pte.hu; 2Department of Pharmacology and Pharmacotherapy, Medical School, University of Pecs, H-7624 Pecs, Hungary; cseko.kata@pte.hu (K.C.); kemeny.agnes@pte.hu (Á.K.); lilla.draskoczi@gmail.com (L.D.); zsuzsanna.helyes@aok.pte.hu (Z.H.); 3Szentágothai Research Centre, University of Pecs, H-7624 Pecs, Hungary; 4Department of Medical Biology and Central Electron Microscope Laboratory, Medical School, University of Pecs, H-7624 Pecs, Hungary; 5Department of Pathology, Medical School, University of Pecs, H-7624 Pecs, Hungary; kereskai.laszlo@pte.hu; 6Department of Medical Microbiology and Immunology, Medical School, University of Pecs, H-7624 Pecs, Hungary; kocsis.bela@pte.hu; 7Institute of Pharmacognosy, Faculty of Pharmacy, Semmelweis University, H-1085 Budapest, Hungary; boszormenyi.andrea@pharma.semmelweis-univ.hu; 8PharmInVivo Ltd., H-7629 Pecs, Hungary

**Keywords:** clove, Scots pine, essential oil, endotoxin, airway inflammation, mouse model, airway hyperresponsiveness, myeloperoxidase activity, perivascular edema, cytokine

## Abstract

Scots pine (SO) and clove (CO) essential oils (EOs) are commonly used by inhalation, and their main components are shown to reduce inflammatory mediator production. The aim of our research was to investigate the chemical composition of commercially available SO and CO by gas chromatography–mass spectrometry and study their effects on airway functions and inflammation in an acute pneumonitis mouse model. Inflammation was evoked by intratracheal endotoxin and EOs were inhaled three times during the 24 h experimental period. Respiratory function was analyzed by unrestrained whole-body plethysmography, lung inflammation by semiquantitative histopathological scoring, myeloperoxidase (MPO) activity and cytokine measurements. α-Pinene (39.4%) was the main component in SO, and eugenol (88.6%) in CO. Both SO and CO significantly reduced airway hyperresponsiveness, and prevented peak expiratory flow, tidal volume increases and perivascular edema formation. Meanwhile, inflammatory cell infiltration was not remarkably affected. In contrast, MPO activity and several inflammatory cytokines (IL-1β, KC, MCP-1, MIP-2, TNF-α) were aggravated by both EOs. This is the first evidence that SO and CO inhalation improve airway function, but enhance certain inflammatory parameters. These results suggest that these EOs should be used with caution in cases of inflammation-associated respiratory diseases.

## 1. Introduction

Essential oils (EOs) are volatile liquids containing mainly terpenoids (mono-, sesqui- and diterpenes) and phenylpropane derivatives, which completely evaporate at room temperature. They can be extracted by distillation from various plant parts and are becoming more and more popular in different fields, e.g., aromatherapy, food preservation, plant pathology and medical microbiology. Inflammatory lung diseases are among the leading causes of death worldwide [1] and they can substantially deteriorate patients’ quality of life. Due to the volatile character of EOs, they can enter the respiratory tract via inhalation. Therefore, patients willingly apply EOs as a single or adjuvant treatment for respiratory diseases and inflammations based on traditional use and empirical evidence. Another favorable characteristic of EOs is that by containing several components they can exert pleiotropic effects through a broad range of targets, such as lipoxygenase, cyclooxygenase and inducible nitric oxide synthase, as well as different receptors, transporters and ion channels [2,3]. The anti-inflammatory effects of eucalyptus, lavender, thyme and cinnamon EOs, 1,8-cineole (eucalyptol), and menthol in the airways have already been proved by our research group [4], as well as others [2,5,6,7].

Scots pine (SO) and clove (CO) EOs are among the most popular oils and are commonly used via inhalation. Their main components, α-pinene and eugenol, have been shown to reduce inflammatory cytokine production and some inflammatory parameters [8,9,10,11,12]. However, these previous in vitro results describe the effect of only one EO component and focus only on certain selected inflammatory parameters; therefore, they are not conclusive and difficult to compare. There are only few in vivo data available, especially on airway function analysis, data which would be important to determine their potential clinical benefits and risks, particularly under inflammatory conditions. 

SO is obtained by steam distillation of the fresh needles (leaves) and twigs of *Pinus sylvestris* L. (Pinaceae). The traditional uses of SO include treating respiratory infections based on its antibacterial, anti-inflammatory, expectorant and analgesic potentials [13]. CO is obtained by the same method from the dried flower buds of *Syzygium aromaticum* (L.) Merill et L. M. Perry (Myrtaceae). According to the European Medicines Agency monograph on clove oil, it is a traditional herbal medicinal product for the symptomatic treatment of oral cavity or pharyngeal minor inflammations and for the temporary relief of toothache due to cavity [14,15]. We selected these two EOs based on their frequent use and presence in different traditional herbal medicinal products administered by inhalation.

The endotoxin (lipopolysaccharide: LPS)-induced acute airway inflammation mouse model is the most frequently applied, well-reproducible mechanism model for the study of EOs in acute respiratory inflammation [4,16,17,18,19]. LPS is a component of the Gram-negative bacterial cell wall, which induces interstitial pneumonitis and acute respiratory obstruction by a well-defined Toll-like receptor-4 activation on macrophages, resulting in the release of several inflammatory mediators and consequent neutrophil activation [20]. 

In our experiments, we aimed to analyze the chemical composition of SO and CO, and perform a complex investigation on their effects related to airway functions and a range of inflammatory parameters in the endotoxin-induced acute pneumonitis mouse model. 

## 2. Results

### 2.1. Chemical Analysis of EOs

The chemical composition of commercially available SO and CO EOs was measured by gas chromatography with a flame-ionization detector (GC-FID) and mass spectrometry (GC-MS). The main components in SO and CO were *α*-pinene (39.4%) and eugenol (88.6%), respectively (Table 1). In SO, limonene (14.3%), *β*-pinene (11.0%), *α*-terpineol (8.8%), *β*-caryophyllene (8.4%), *δ*-3-carene (7.0%), bornyl acetate (3.4%), and fenchone (1.1%) were also present, while in CO, *β*-caryophyllene (8.6%), and *α*-humulene (2.2%) were present in higher concentrations.

### 2.2. Respiratory Functions

LPS treatment significantly reduced breathing frequency and minute ventilation, and increased tidal volume, time of inspiration and expiration, and peak expiratory flow 24 h after administration. Meanwhile, it did not alter peak inspiratory flow and relaxation time (Figure 1a–h). Both EOs significantly alleviated carbachol-induced airway hyperresponsiveness and inhibited LPS-induced peak expiratory flow increase (Figure 1h,i). SO inhalation also significantly reduced tidal volume (Figure 1b) compared to the LPS-PO-treated group, but had no other observable effect on other airway parameters. 

### 2.3. Lung Histopathological Evaluation

LPS administration induced neutrophil granulocyte and macrophage infiltration associated with a remarkable perivascular and peribronchial edema formation (Figure 2 and Figure 3). Neither SO nor CO significantly altered any LPS-induced histopathological parameters, partially due to the great interindividual variation of the semiquantitative score values. However, in contrast to the PO-treated controls, none of the observed inflammatory markers were significantly increased in the CO-treated group, and edema as well as macrophage infiltration did not significantly elevate in the SO-treated group either, showing anti-inflammatory actions of the investigated EOs (Figure 3).

### 2.4. LPS-Induced Lung Myeloperoxidase (MPO) Activity Was Aggravated by SO and CO Inhalation

MPO production correlating with granulocyte and macrophage activity was measured from the lung homogenates. LPS treatment significantly increased MPO activity, and surprisingly both SO and CO treatments induced an approximately 2-fold further elevation of this inflammatory parameter (Figure 4).

### 2.5. SO and CO Aggravated LPS-Evoked Inflammatory Cytokine Concentrations of the Lung

SO significantly aggravated LPS-induced interleukin-1beta (IL-1β), keratinocyte chemoattractant (KC), monocyte chemoattractant protein 1 (MCP-1), macrophage inflammatory protein 2 (MIP-2), and tumor necrosis factor alpha (TNF-α), while CO significantly enhanced interleukin-1 (IL-6), and KC (Figure 5).

## 3. Discussion

Since the in vivo effects of SO and CO EOs are not convincingly demonstrated in the literature, and the data are partially contradictory, we therefore focused on testing the effect of these EOs in an endotoxin-induced acute pneumonitis mouse model. Our study provides here the first evidence that both SO and CO (with α-pinene and eugenol as main components, respectively) inhalation decrease inflammatory airway hyperresponsiveness in the LPS-induced acute lung injury model. Neither EO induced histopathological changes in the non-inflamed lung, but in the case of both treatments, LPS-induced characteristic inflammatory alterations as shown by the semiquantitative scores were not significant. Surprisingly, in contrast to these functional and morphological results, MPO activity and several inflammatory cytokines were remarkably aggravated by both EOs.

The novelty of our work is emphasized by the fact that this is the first study demonstrating the in vivo effects of SO on inflammation. Our findings are partially consistent with previous results obtained with components of SO and EOs of other pine species.

Orally administered maritime pine (*Pinus pinaster* Ait) EO (with 13.5% α-pinene content) did not show anti-inflammatory effects in the carrageenan-induced hind paw edema mouse model [21].

Myrtol, a standardized mixture of EOs primarily containing cineole, limonene and α-pinene (from Pinus species) administered orally, inhibited LPS-induced neutrophil accumulation, TNF-α and IL-6 concentrations, but, in agreement with our results, increased MPO activity 6 h after LPS administration in the mouse [22]. These partially contradictory data might be due to the different EO composition and the duration of the study. In a clinical study, Myrtol improved mucociliary clearance in patients with chronic obstructive pulmonary disease, but did not affect respiratory functions [23]. Short-term exposure of α-pinene did not evoke acute lung function changes, but caused mild irritation in higher concentrations [24]. However, the inhalation of turpentine with high (35%) δ-3-carene content increased airway resistance, besides causing discomfort in the throat and airways of healthy men [25]. 

α-Pinene, the main component of SO, decreased TNF-α, IL-1β, IL-6, intercellular adhesion molecule-1 (ICAM), and macrophage inflammatory protein-2 (MIP-2) levels, as well as eosinophil and mast cell infiltration in the nasal mucosa in an ovalbumin-sensitized allergic rhinitis mouse model [26]. It inhibited IL-6, TNF-α and NO productions in LPS-activated isolated mouse peritoneal macrophages by suppressing the mitogen-activated protein kinases (MAPKs) and the nuclear factor-kappa B (NF-κB) pathways [11]. EOs obtained from *P. heldreichii* (α-pinene 10.57%), *P. peuce* (α-pinene 36.79%) and *P. mugo* (α-pinene 21.34%) reduced IL-6 secretion from the LPS-activated RAW 264.7 mouse monocyte/macrophage-like cell line only in higher concentrations, but did not modify or even increased secretion in lower concentrations [27], suggesting clear concentration-dependent effects.

Meanwhile, *P. pinaster* EO (α-pinene 62%) did not inhibit LPS-indued TNF-α and CCL2 production in human acute monocytic leukemia cells (THP-1) [28]. 

Regarding *Syzygium aromaticum*, it is important to note that most available data refer to the aqueous extract but not the EO of the plant, or focused only on the main component, eugenol. The aqueous extract of clove (i.p.) significantly reduced matrix metalloproteinase-2 (MMP-2) and -9 activities, neutrophil count and protein leakage into bronchoalveolar lavage fluid in an LPS-induced inflammation mouse model. Furthermore, it also decreased MPO activity concentration-dependently in phorbol myristate acetate-stimulated human neutrophils [29]. The same extract reduced carrageenan-induced hind paw edema and liver succinate dehydrogenase and xanthine oxidase activities in the rat [30]. 

CO did not affect phagocytosis. It significantly increased the expression of inducible nitric oxide synthase (iNOS), but reduced IL-6 production in LPS-activated RAW 264.7 cells [31].

Eugenol, the main component of CO, is the most thoroughly studied substance in relation to inflammation [8,9,10,12]. Eugenol did not affect LPS-induced respiratory changes, but reduced lung edema, inflammatory cells, and IL-6 and IL-1β levels in the bronchoalveolar lavage fluid, as well as inflammatory cell infiltration [32]. 

In another study, i.p. eugenol reduced LPS-induced pulmonary inflammation, improving lung function as well as significantly reducing neutrophil and macrophage counts and TNF-α level [33]. Similar to this finding, eugenol inhibited changes in lung mechanics, pulmonary inflammation, and alveolar collapse elicited by diesel particles [34]. In contrast to these experimental data, eugenol may cause different adverse reactions in humans, including skin irritation, inflammation, ulcer formation, dermatitis, or slow healing [35]. 

These controversial findings with both SO and CO are likely to be due to the different compositions of the aqueous extracts and the EOs, the different effects of the main components by themselves as compared to the complexes, the differences between the ways of administration and concentration, and the mechanisms of the inflammatory processes. Despite the fact that our results provide useful information regarding the potential benefits and risks of EO inhalation in airway inflammation, a limitation of our experimental design is that we could not measure the exact EO concentration in the inhalation box, but could only calculate its maximum value.

## 4. Materials and Methods

### 4.1. EO Samples and the Gas Chromatographic Analysis of Their Composition

Scots pine (*Pinus sylvestris* L.) and clove (*Syzygium aromaticum* (L.) Merr. & L. M. Perry syn. *Eugenia caryophyllata* Thunb.) EOs were bought from Aromax Ltd. (Budapest, Hungary). To analyze the chemical composition of the EO samples, gas chromatography–mass spectrometry (GC-MS, Agilent 6890 N/5973 N GC-MSD, Santa Clara, CA, USA) was used. The percentage compositions of the EOs were evaluated by a flame ionization detector (FID). Compounds were identified based on retention data and spectral data of standard compounds, and the NIST 05 mass spectral library was also applied as previously described [36,37].

### 4.2. Animals

In the animal experiments, 10–18-week-old female C57BL/6J mice [20] weighing 21.7 ± 0.30 g (mean ± SEM) at the beginning of the experiment were used. The age distribution of the mice was similar in each group to avoid age-related differences. They were bred and kept in the Laboratory Animal House of the Department of Pharmacology and Pharmacotherapy at the University of Pécs. Optimal parameters were provided for all the animals (e.g., 325 × 170 × 140 mm cages, 12 h light/dark cycle, 24–25 °C, mouse chow, water).

During the experiments, the following regulations were considered: European legislation (Directive 2010/63/EU) and Hungarian Government regulation (40/2013., II. 14.) on the protection of animals used for scientific purposes, and the recommendations of the International Association for the Study of Pain. The study design was approved by the Ethics Committee on Animal Research of the University of Pécs (license No.: BA02/2000-26/2018, 21 June 2018).

### 4.3. Induction of Acute Airway Inflammation and Groups of Animals

Animals received 100 µg *Escherichia coli* (serotype: 083) LPS intratracheal (i.t.) dissolved in 60 µL phosphate-buffered saline (PBS) to induce acute airway inflammation. The endotoxin was isolated and purified in the Department of Microbiology, University of Pécs. The animals were put under ketamine (100 mg/kg i.p.; Sigma Aldrich, St. Louis, MO, USA) and xylazine (5 mg/kg i.p.; Sigma Aldrich, St. Louis, MO, USA) anesthesia during the administration of LPS. Control mice received the same volume of sterile PBS [38]. Animals were treated with 30 min EO inhalation 1 h prior to and 4 and 23 h following LPS/PBS administration as previously described [4]. The negative control of the EOs was paraffin oil (PO). The maximum concentration of the EOs was calculated as 6.55 µL/L. Mice were randomized into six groups: (1) the control group treated with PBS i.t. and PO inhalation, (2) mice treated with LPS i.t. and PO inhalation, (3) PBS i.t. and SO inhalation, (4) LPS i.t. and SO inhalation, (5) PBS i.t. and CO inhalation, (6) LPS i.t. and CO inhalation (*n* = 6–8/group). 

### 4.4. Pulmonary Function Measurement

Respiratory functions were determined in conscious and spontaneously breathing animals by unrestrained whole-body plethysmography (WBP) (PLY3211, Buxco Europe Ltd., Winchester, UK) 24 h after PBS/LPS administration [39]. Baseline measurements were registered with aerosolized saline to determine the respiratory parameters, such as breathing frequency, tidal volume, minute ventilation, relaxation time, time of inspiration and expiration, peak inspiratory and expiratory flow. The enhanced pause (Penh; calculated as: (expiratory time/relaxation time)/(max. expiratory flow/max. inspiratory flow)) correlating with airway hyperresponsiveness [39] was measured after carbachol (11 and 22 mm; Sigma Aldrich, St. Louis, MO, USA)-induced bronchoconstriction as previously described [4]. Airway function parameters were registered and averaged by the BioSystem XA Software for Windows (Buxco Research Systems, Wilmington, NC, USA).

### 4.5. Histopathological Assessment and Semiquantitative Scoring

After airway function measurements, the mice were anaesthetized, and their lungs were harvested. Left lungs were fixed in 4% formaldehyde, embedded in paraffin, sectioned with a microtome (5–7 μm), and stained with hematoxylin–eosin. Semiquantitative histopathological scoring was performed by an expert pathologist in a blinded manner. Perivascular edema (0–3), perivascular/peribronchial neutrophil accumulation (0–3), and infiltration of macrophages/mononuclear cells in the alveolar spaces (0–2) were scored as previously described [40,41]. The total inflammatory score was assessed (0–8) by adding the subscores of the individual histopathological parameters. 

### 4.6. Spectrophotometric Measurement of Myeloperoxidase (MPO) Activity

Lung inflammation was further characterized by MPO enzyme activity assessment correlating with activated neutrophil and macrophage infiltration. Spectrophotometric measurement was performed from the right lung homogenates using H_2_O_2_-3,3′,5,5′-tetramethylbenzidine (TMB/H_2_O_2_), and MPO activity was compared to a human standard MPO preparation as described earlier [39]. All reagents were obtained from Sigma-Aldrich Ltd. (St. Louis, MO, USA).

### 4.7. Measurement of Inflammatory Cytokine Concentration Using Luminex xMAP Technology 

The excised and frozen lung tissue samples were thawed and weighed, then immediately placed in cold PBS containing 0.01% phenylmethanesulfonyl fluoride (PMSF, Sigma Aldrich, St. Louis, MO, USA, P7626) protease inhibitor, and were homogenized on ice with a Miccra D-9 Digitronic device (Art-moderne Laborteknik, Germany). Homogenates were centrifuged for 20 min (4 °C, 4000 rpm) and clear supernatants were collected and stored at −80 °C until the measurement. With a customized Milliplex Mouse Cytokin/Chemokine Magnetic Bead Panel (MCYTOMAG-70K), Luminex Multiplex Immunoassay was performed to determine the protein concentrations of the following cytokines/chemokines: interleukin-1beta (IL-1β); interleukin-6 (IL-6); chemokine (C-X-C motif) ligand 1 (CXCL1), also called keratinocyte chemoattractant (KC); chemokine (C-C motif) ligand 2 (CCL2), also called monocyte chemoattractant protein 1 (MCP-1); chemokine (C-X-C motif) ligand 2 (CXCL2), also called macrophage inflammatory protein 2 (MIP-2); and tumor necrosis factor alpha (TNF-α). The six analytes were detected simultaneously in a 96-well plate. All samples were tested undiluted in a blind fashion and in duplicate. Based on the instructions, a mixture of the six antibody-coated bead population was added to the plate at 25 µL/well together with standards and controls to the designated wells. Following overnight incubation at 4 °C, the plate was washed three times using a handheld magnetic plate. After washing, 25 µL/well of detection antibody solution was added and incubated for 60 min at RT with shaking at 500 rpm. After subsequent washing steps, 25 µL/well of streptavidin–phycoerythrin (SAPE) solution was added and incubated for 30 min at RT shaking at 500 rpm. After washing three times, 150 µL/well of drive buffer was added to the plate and the assay was read on a MAGPIX Luminex (TermoFisher, Budapest, Hungary) instrument. Five-PL regression curves were used to plot the standard curves for all analytes by the Belysa 1.1 software (Sigma Aldrich, St. Louis, MO, USA) analyzing the bead median fluorescence intensity. Results are given in pg/mL and were normalized to the total protein concentration of the sample.

### 4.8. Statistical Analysis of Data

Statistical analysis was performed in GraphPad Prism v6 (GraphPad Software, San Diego, CA, USA). Unless noted otherwise, all data represent the mean ± SEM. Respiratory parameters and MPO activity were analysed with two-way ANOVA followed by Tukey post-test. Composite histopathological inflammatory score values were evaluated by Kruskal–Wallis analysis followed by Dunn’s post-test. In all cases, *p* < 0.05 was accepted as significant. 

## 5. Conclusions

This study provides the first evidence that SO and CO inhalation improve airway function, but enhance certain inflammatory parameters. Therefore, we conclude that these EOs could be beneficial for certain functional respiratory disorders, but should be used with caution in cases of inflammation-associated airway conditions. 

## Figures and Tables

**Figure 1 molecules-27-03868-f001:**
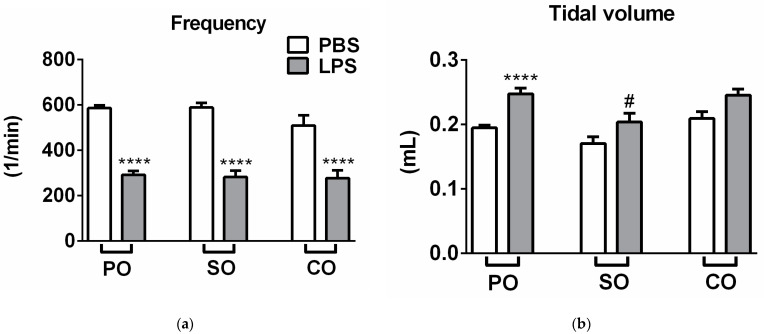
Effects of Scots pine (SO) and clove (CO) essential oils on (**a**) breathing frequency, (**b**) tidal volume, (**c**) minute ventilation, (**d**) relaxation time, (**e**) time of inspiration, (**f**) time of expiration, (**g**) peak inspiratory flow, (**h**) peak expiratory flow and (**i**) Penh compared to the negative control paraffin oil (PO), after lipopolysaccharide (LPS-black columns)/phosphate-buffered saline (PBS-white columns) treatment (*n* = 6–8/group, * *p* < 0.05, ** *p* < 0.005, *** *p* < 0.0005, **** *p* < 0.0001 vs. respective PBS-treated group, # *p* < 0.05, ## *p* < 0.005, ### *p* < 0.0005 vs. LPS-PO-treated mice; two-way ANOVA followed by Tukey post-test).

**Figure 2 molecules-27-03868-f002:**
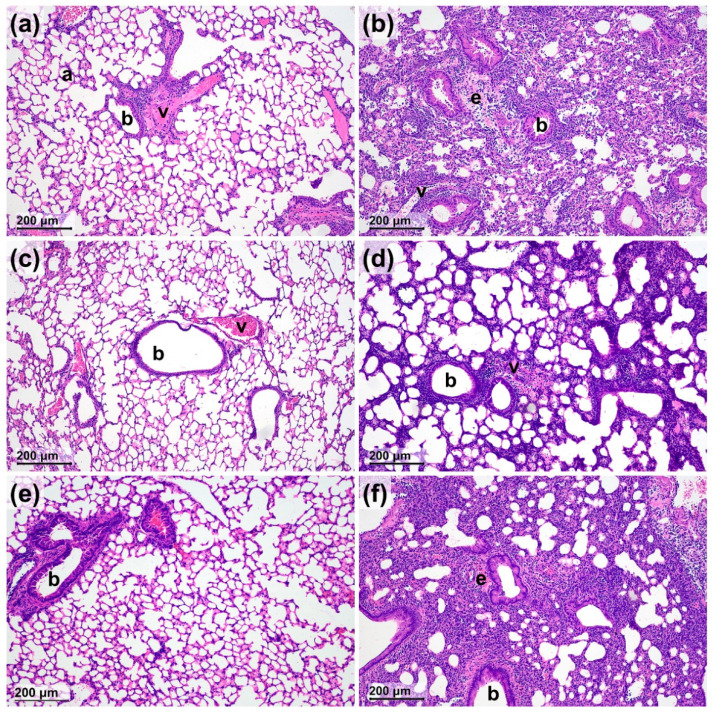
Representative histopathological pictures of non-inflamed PBS-treated and LPS-treated inflamed lung parenchyma of mice after Scots pine (SO) and clove oil (CO) inhalation in comparison with the control paraffin oil (PO). (**a**) PBS and PO treatment, (**b**) LPS and PO treatment, (**c**) PBS and SO treatment, (**d**) LPS and SO treatment, (**e**) PBS and CO treatment, and (**f**) LPS and CO treatment (hematoxylin–eosin staining, 100× magnification; a: alveolus, b: bronchiole, v: vessel, e: edema).

**Figure 3 molecules-27-03868-f003:**
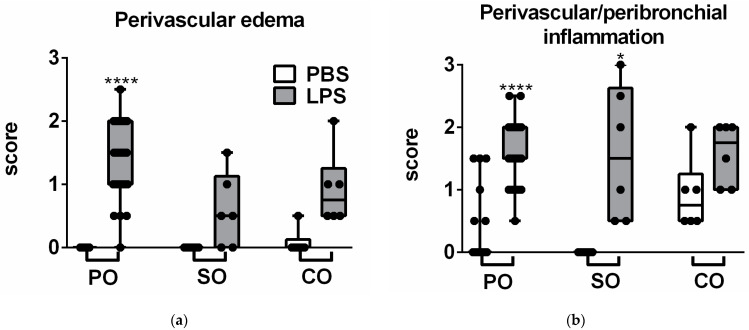
Semiquantitative evaluation of (**a**) perivascular and peribronchial edema, (**b**) accumulation of neutrophil granulocytes, (**c**) macrophages and (**d**) total score of lung histopathological alterations. Individual data points (*n* = 6–8 mice/treatment groups and 29 in the control groups) are demonstrated with the medians and the SEM (* *p* < 0.05, **** *p* < 0.0001 vs. respective PBS-treated mice; Kruskal–Wallis analysis followed by Dunn’s post-test).

**Figure 4 molecules-27-03868-f004:**
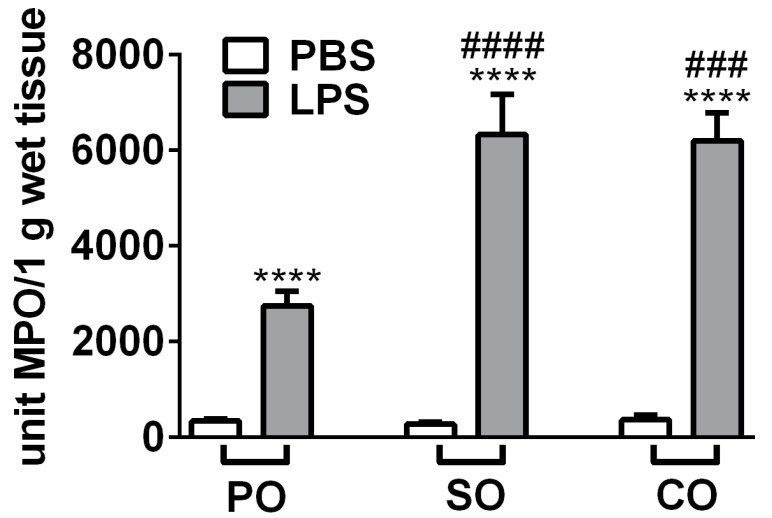
Myeloperoxidase (MPO) activity of the lung homogenates. Columns represent the means ± SEM of *n* = 6–8 mice/group (**** *p* < 0.0001 vs. respective PBS-treated group, ### *p* < 0.0005, #### *p* < 0.0001 vs. LPS-PO-treated mice; two-way ANOVA followed by Tukey post-test).

**Figure 5 molecules-27-03868-f005:**
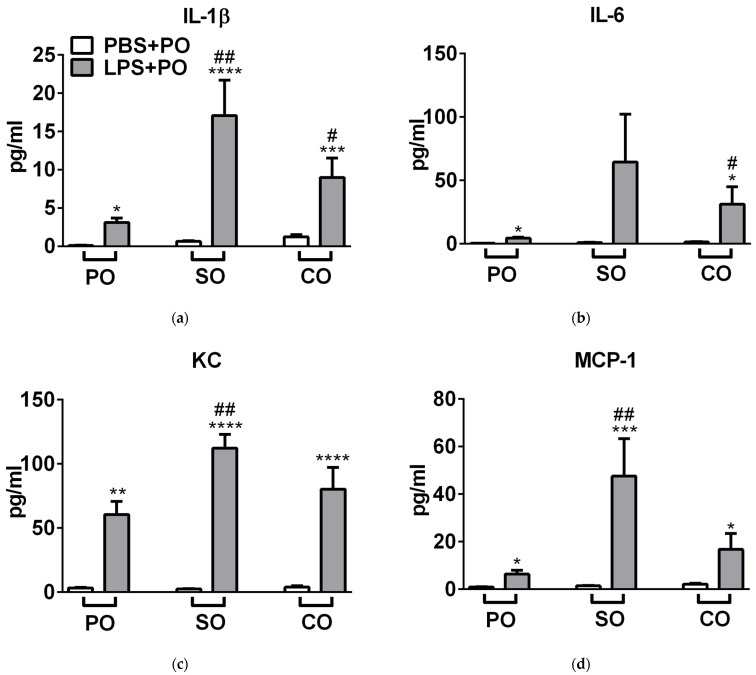
Cytokine levels of the lung homogenates. Columns represent the means ± SEM of *n* = 6–8 mice/group (* *p* < 0.05, ** *p* < 0.005, *** *p* < 0.0005, **** *p* < 0.0001 vs. respective PBS-treated group, # *p* < 0.05, ## *p* < 0.005 vs. LPS-PO-treated mice; two-way ANOVA followed by Tukey post-test).

**Table 1 molecules-27-03868-t001:** The chemical composition of the investigated essential oils in percentage from 3 parallel measurements (RI: retention index).

Compound	RI	Scots Pine (%)	Clove (%)
α-Pinene	959	39.4	-
Camphene	977	0.8	-
Fenchone	982	1.1	-
*β*-Pinene	1005	11.0	-
δ-3-Carene	1036	7.0	-
Limonene	1055	14.3	-
α-Terpineol	1216	8.8	-
Bornyl acetate	1300	3.4	-
Eugenol	1372	-	88.6
Longifolene	1429	0.5	-
*β*-Caryophyllene	1435	8.4	8.6
α-Humulene	1471	0.8	2.2
Caryophyllene oxide	1594	0.7	0.5
Total		96.2	99.9

## Data Availability

The data presented in this study are available on request from the corresponding author.

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
