# Peer review of "Pinus sylvestris L. and Syzygium aromaticum (L.) Merr. & L. M. Perry Essential Oils Inhibit Endotoxin-Induced Airway Hyperreactivity despite Aggravated Inflammatory Mechanisms in Mice"

_molecules, 2022, doi:10.3390/molecules27123868_

Round 1
Reviewer 1 Report
abst l24... .. commonly used for inhalation.. should be changed to: ..used by inhalation..
lines 25-6"However, their 25 in vivo effects are not convincingly demonstrated, and data are partially contradictory. Therefore, 26 we.."
should be removed but can be used in the discussion part.
rephrase the the aim of the work.
as this mans was submitted for the special issue on essential oils, there is no need to define essential oils in the intro part. Remove Lines44-48 sentence. As this is not reflecting also reference 1.
The number of references should be revised,only neccessary ones should be kept. All plant species must be in italics wherever applicable.
highlight clear the objective and aim of the work in the intro part. Please also explain why in particular these two oils were selected. also explain why the major constituents were not also tested / evaluated?.. there is already some discussion part, however, it can be improved.
what was used as positive and negative control in the assays?
After revisions the work would be certainly interest for the aromatherapy audience and can be considered for publication upon the decision of the editorial.
Author Response
Dear Reviewer 1,
First of all, we are highly grateful for the valuable comments. As a correspondence author I would like to submit our answer for Reviewer 1 related to our revised manuscript entitled „Pinus sylvestris L. and Syzygium aromaticum (L.) Merr. & L.M.Perry essential oils inhibit endotoxin-induced airway hyperreactivity despite aggravated inflammatory mechanisms in mice” for your consideration to be published in special issue entitled "Antibacterial and Biological Activity of Essential Oils: From In Vitro to In Vivo Evidence" of Molecules.
Comment of Reviewer 1:
abst l24... .. commonly used for inhalation.. should be changed to: ..used by inhalation.. – Thank you for this recommendation, we corrected it.
lines 25-6 "However, their 25 in vivo effects are not convincingly demonstrated, and data are partially contradictory. Therefore, 26 we..." should be removed but can be used in the discussion part. – We removed this part from the abstract and included into discussion part as you suggested.
rephrase the the aim of the work. – Thank you for this comment, we rephrased it.
as this mans was submitted for the special issue on essential oils, there is no need to define essential oils in the intro part. Remove Lines44-48 sentence. As this is not reflecting also reference 1. – Thanks for this recommendation. However, if somebody makes a general search for our manuscript from any databases, e.g. PubMed, Sciencedirect, Google Scholar, etc. the definition of EO is necessary. Therefore, we would like to keep it in the introduction part. Ref 1 focuses on the frequency of lung diseases not the EOs, therefore, we would like to include it into our references.
The number of references should be revised, only neccessary ones should be kept. All plant species must be in italics wherever applicable. – Thank you for this comment, we used italics for the names of all plant species.
highlight clear the objective and aim of the work in the intro part. Please also explain why in particular these two oils were selected. also explain why the major constituents were not also tested / evaluated?.. there is already some discussion part, however, it can be improved. - We highlighted the objective and the aim of our work in the introduction part. Furthermore, we explained why these two EOs were used. We did not test the major compounds because there are several in vivo evidences about the effect of alpha-pinene and eugenol as we described in the discussion part.
what was used as positive and negative control in the assays? – In our study paraffin oil (PO) was used as negative control, as we indicated in the Materials and Methods section of the revised version of the manuscript. The LPS-induced acute lung injury model is validated with the use of the reference compound dexamethason as a positive control. Some previous references focused on it:
Pinheiro, A., Mendes, A., Neves, M., Prado, C. M., Bittencourt-Mernak, M. I., Santana, F., Lago, J., de Sá, J. C., da Rocha, C. Q., de Sousa, E. M., Fontes, V. C., Grisoto, M., Falcai, A., & Lima-Neto, L. G. (2019). Galloyl-Hexahydroxydiphenoyl (HHDP)-Glucose Isolated From Punica granatum L. Leaves Protects Against Lipopolysaccharide (LPS)-Induced Acute Lung Injury in BALB/c Mice. Frontiers in immunology, 10, 1978. https://doi.org/10.3389/fimmu.2019.01978
Lin, Y., Zhang, M., Lu, Q., Xie, J., Wu, J., & Chen, C. (2019). A novel chalcone derivative exerts anti-inflammatory and anti-oxidant effects after acute lung injury. Aging, 11(18), 7805–7816. https://doi.org/10.18632/aging.102288
Huang, X., Zhu, J., Jiang, Y., Xu, C., Lv, Q., Yu, D., Shi, K., Ruan, Z., & Wang, Y. (2019). SU5416 attenuated lipopolysaccharide-induced acute lung injury in mice by modulating properties of vascular endothelial cells. Drug design, development and therapy, 13, 1763–1772. https://doi.org/10.2147/DDDT.S188858
Li, W. W., Wang, T. Y., Cao, B., Liu, B., Rong, Y. M., Wang, J. J., Wei, F., Wei, L. Q., Chen, H., & Liu, Y. X. (2019). Synergistic protection of matrine and lycopene against lipopolysaccharide‑induced acute lung injury in mice. Molecular medicine reports, 20(1), 455–462. https://doi.org/10.3892/mmr.2019.10278
Hussain, M., Xu, C., Wu, X., Lu, M., Tang, L., Wu, F., Wu, X., & Wu, J. (2019). A CRTH2 antagonist, CT-133, suppresses NF-κB signalling to relieve lipopolysaccharide-induced acute lung injury. European journal of pharmacology, 854, 79–91. https://doi.org/10.1016/j.ejphar.2019.03.053
In our study we did not use a reference compound (dexamethasone), as our aim was not to assess whether essential oil can achieve or surpass the effect of therapeutic agents, but to determine their potential as adjunct treatment in airway inflammation.
After revisions the work would be certainly interest for the aromatherapy audience and can be considered for publication upon the decision of the editorial.
Reviewer 2 Report
The manuscript is on the analysis and some pharmacological activities of two commercially purchased scots pine and clove essential oils. The analyses prove the correctness of the essential oils used in the study. I suggested the authors give retention indices of the compounds instead of retention times since retention indices are more commonly accepted values for the correct characterization of essential oil constituents than retention times. There is no compound called longiferene, and I asked if it could be a misspelling of longifolene. If these are corrected analysis of the essential oil part of the manuscript will be completed. I did not make any comment on the pharmacological part of the manuscript since I do not feel fully qualified to comment on that issue.
Retention indices instead of retention times must be indicated.
Longiferene does not exist. Could it be longifolene?
Author Response
Dear Reviewer 2,
First of all, we are highly grateful for the valuable comments. As a correspondence author I would like to submit our answer for Reviewer 2 related to our revised manuscript entitled „Pinus sylvestris L. and Syzygium aromaticum (L.) Merr. & L.M.Perry essential oils inhibit endotoxin-induced airway hyperreactivity despite aggravated inflammatory mechanisms in mice” for your consideration to be published in special issue entitled "Antibacterial and Biological Activity of Essential Oils: From In Vitro to In Vivo Evidence" of Molecules.
Comment of Reviewer 2:
The manuscript is on the analysis and some pharmacological activities of two commercially purchased scots pine and clove essential oils. The analyses prove the correctness of the essential oils used in the study. I suggested the authors give retention indices of the compounds instead of retention times since retention indices are more commonly accepted values for the correct characterization of essential oil constituents than retention times. There is no compound called longiferene, and I asked if it could be a misspelling of longifolene. If these are corrected analysis of the essential oil part of the manuscript will be completed. I did not make any comment on the pharmacological part of the manuscript since I do not feel fully qualified to comment on that issue.
Retention indices instead of retention times must be indicated. – Thank you for this comment, we changed retention times to retention indices.
Longiferene does not exist. Could it be longifolene? – It was a mispelling, thank you, we corrected it.